# PlasticityNet: Learning to Simulate Metal, Sand, and Snow for Optimization Time Integration

**Xuan Li**
Department of Mathematics
University of California, Los Angeles
xuanli1@math.ucla.edu

**Yadi Cao**
Department of Computer Science
University of California, Los Angeles
yadicao95@cs.ucla.edu

**Minchen Li**
Department of Mathematics
University of California, Los Angeles
minchen@math.ucla.edu

**Yin Yang**
School of Computing
University of Utah
yin.yang@utah.edu

**Craig Schroeder**
Department of Computer Science and Engineering
University of California, Riverside
craigs@cs.ucr.edu

**Chenfanfu Jiang**
Department of Mathematics
University of California, Los Angeles
cffjiang@math.ucla.edu

## Abstract

In this paper, we propose a neural network-based approach for learning to represent the behavior of plastic solid materials ranging from rubber and metal to sand and snow. Unlike elastic forces such as spring forces, these plastic forces do not result from the positional gradient of any potential energy, imposing great challenges on the stability and flexibility of their simulation. Our method effectively resolves this issue by learning a generalizable plastic energy whose derivative closely matches the analytical behavior of plastic forces. Our method, for the first time, enables the simulation of a wide range of arbitrary elasticity-plasticity combinations using time step-independent, unconditionally stable optimization-based time integrators. We demonstrate the efficacy of our method by learning and producing challenging 2D and 3D effects of metal, sand, and snow with complex dynamics.

## 1 Introduction

Combining machine learning with physical simulations has recently attracted a lot of attention. A vast amount of existing research adopts an end-to-end approach, where the specific underlying computational physics system is treated as a black box [46, 41]. Harnessing the power of neural networks, this research has been successfully applied in computer animation [10], multibody systems [3, 6, 59, 12], human musculature simulation [20], computational fluid dynamics [4, 13], and non-linear continuum mechanics [5]. An alternative direction is represented by physics-informed neural networks (PINN) [44, 21], where in its original form, the residual of a partial differential equation is directly used as the loss function so that the network training is a physics-aware learning process. PINN becomes powerful when the design space of the input to the network can be parameterized, which accelerates both the roll-out and the inverse optimization process [51]. Another noteworthy category is learning the physical modeling where the machine can either help increase the model resolution in a coarser grid [24], inject nonlinearity to a linear model [37], or apply a learnable model reduction to reduce the system degrees-of-freedom (DOF) for acceleration [47, 48].

36th Conference on Neural Information Processing Systems (NeurIPS 2022).

Despite its great success, training a neural network to replace a traditional simulator is not always the preferred choice. This is partially due to the challenges in the trained model's generality and portability. For example, a trained model on a particle-based deformable body solver (such as the Material Point Method (MPM [19]) cannot be directly applied to the mesh-based Finite Element Method (FEM) [49], while in traditional continuum mechanics, the constitutive model that describes the relationship between force and deformation is an independent module from the underlying geometric description or simulation scheme. Indeed, by simply switching the constitutive model and applying minor changes to the existing and general simulation pipeline, a wide range of materials can be simulated in the same framework, ranging from sand [42, 9, 23, 52] and metal [39], to snow [15, 57, 35] and glacier [58].

Many elastic materials, including those represented by mass-spring systems [2] and common *hyperelastic* solids [50], are usually governed by analytical elastic potential energy functions in terms of the deformation. These models are well fitted to experiments and proven to be simple, accurate, and predictive. Although most of these energy functions are highly nonlinear and non-convex, reformulating the dynamic simulation process as a numerical optimization problem and solving it using projected Newton and line search can guarantee global convergence to a solution [28]. Beyond hyperelasticity, plasticity is much more challenging. The mechanical response of plastic materials imposes extra difficulties in the implementation as it is path-dependant and non-smooth. One common handling of plasticity is the return mapping algorithm, which applies the effects of plastic deformation to the elastic forces. However, this leads to asymmetrical force derivatives, which eliminate the possibility of integrating the plasticity into the energy function in a single optimization and complicates the pipeline. In the recent work of Energetic Consistent Inelasticity (ECI) [34], the plasticity is analytically modeled as an energy functional, and the simulation can be formulated as an optimization problem just like simulating pure elastic materials. However, their analytical derivation only works for St.Venant-Kirchhoff (StVK) elasticity with the von-Mises plasticity.

In this work, we propose PlasticityNet, a neural network-based approach for learning an energy-based force that locally approximates elastic forces with plasticity models and is compatible with optimization time integrators. PlasticityNet framework supports any combinations of elastic models and plastic models and works with both MPM and FEM discretizations. With optimization time integrators, we demonstrate that our framework can simulate vast types of plasticities, such as metal, sand, and snow, with large time step sizes.

## 2   Related work

**Classic Plasticity Models**   The classic plastic models utilized the geometry information of the plasticity and are available for many applications. In the computer graphics community, researchers have followed mechanical literature on the Drucker-Prager elastoplasticity model [42, 9], and developed particle-based simulations of dry [23] and wet [52] sand. Extending a similar Cam-Clay plasticity model, snow avalanches [15, 35], glacier calving [58] and food fracturing [57] are also captured with high visual plausibility as well as physical accuracy. For metals and dough-like materials, the von-Mises plasticity model [39] is usually adopted, while [56, 18] presented its anisotropic extensions. Still, the implementation of these models in modern, optimization-based simulators is cumbersome due to the non-integrable forces. Recently, [34] proposed an elastoplastic energy of von-Mises plasticity under StVK elasticity for optimization time integrator, which can be viewed as a special-case analytical solution to our framework under the same combination of elasticity and plasticity. But our framework works for arbitrary combinations.

**Data-Driven Plasticity Models**   The machine learning approach has been used to find new plastic models using large sets of measurements and parameters, outperforming many long-standing hand-crafted models. The macro-level constitutive relationship is learned from the results of the micro-level simulations [40, 45]. A similar approach is applied in [53] to learn anisotropic hyperelasticity, where additional geometrical information is included in the input. PINN can also be applied in plastic model finding from experimental measurements [1, 54, 25], where the loss includes the stress and Hessian, to infer stress with more accuracy in the implicit simulators. However, there does not exist any prior work, to the best of the author's knowledge, that tried to find variational form for arbitrary plasticity model.

**Optimization Time Integration** The optimization time integrators have advantages in terms of stability under large deformations and large time step sizes. Many of the nonlinear systems of equations that arise from implicit simulation can be integrated to get equivalent optimization problems, which allow robust optimization techniques to be applied. The MPM simulator in this work is based on [14], which formulated the backward Euler time integration with hyperelastic materials as a minimization problem. [30] and [55] also explored domain decompositions and hiearachical preconditioners to improve robustness and efficiency. The FEM simulator in this work is based on Incremental Potential Contact (IPC) [29], which proposed a variational form for frictional contacts. Their optimization-based frictional contact framework was also extended to codimensional objects [31], rigid bodies [11, 26], articulated multibodies [7], reduced elastic solids [27], embedded interfaces [60], and FEM-MPM coupled domains [33].

# 3 Background

## 3.1 Optimization Time Integration

In this section, we briefly introduce the optimization time integration for elastodynamics simulations with the Material Point Method (MPM) and the Finite Element Method (FEM). We refer the readers to [28] and [14] for more details.

FEM discretizes the simulation domain as unstructured meshes (e.g., triangle meshes in 2D), while in MPM, a point cloud composed of material particles is used to discretize the domain. While FEM directly uses the mesh nodes as the simulation degrees-of-freedom (DOF), MPM transfers its particle state to a uniform background grid, whose nodes are used as the DOFs for the integration of forces [19]. Robust simulation of elastodynamics can be achieved via implicit time integration, which updates the nodal positions ($\mathbf{x}$) or velocities ($\mathbf{v}$) step by step based on the previous physical states. To step from $t^n$ to $t^{n+1} = t^n + \Delta t$ with time step size $\Delta t$, with implicit Euler time integration rule, one needs to solve a nonlinear system of equations

$$\mathbf{M}(\mathbf{v}^{n+1} - (\mathbf{v}^n + \mathbf{g}\Delta t)) = \Delta t \mathbf{f}^{n+1}. \tag{1}$$

Here $\mathbf{v}$ is the velocity DOF formed by concatenating all nodal velocity vectors, similarly concatenated, $\mathbf{M}$ is the mass matrix, $\mathbf{g}$ is the gravitational acceleration vector, and $\mathbf{f}$ is the internal force vector. Without plasticity, the internal force on a node $i$ can be calculated as

$$\mathbf{f}_i^{n+1} = -\sum_q V_q^0 \mathbf{P}(\mathbf{F}_q^{n+1}) \nabla w_{iq}, \tag{2}$$

where $q$ iterates the surrounding elements/particles of node $i$ in FEM/MPM, $V_q^0$ is the initial volume of the element/particle, $\mathbf{F} = (\mathbf{I} + \Delta t \nabla \mathbf{v})\mathbf{F}^n$ (MPM) or $\mathbf{F} = \nabla \mathbf{x}^n + \Delta t \nabla \mathbf{v}$ (FEM) is the deformation gradient, which measures deformation from the undeformed state to the deformed state, and $\mathbf{P}$ is the first-Piola Kirchhoff stress, which describes the internal force per unit area within a material. $\nabla w_{iq}$ is the gradient of the weight function on node $i$ evaluated on an element/particle center. The weight function is for transferring physical quantities between the elements/particles and the mesh/grid nodes. Unlike FEM, the last time step is used in MPM as the reference configuration, and so $\nabla w_{iq}$ is calculated as $\mathbf{F}_q^{n\top} \nabla w_{iq}^n$.

When there exists an energy density function $\Psi$ such that $\mathbf{P}(\mathbf{F}) = \frac{\partial \Psi}{\partial \mathbf{F}}$, solving Equation 1 is equivalent to solving the following optimization problem

$$\mathbf{v}^{n+1} = \text{argmin}_\mathbf{v} \frac{1}{2} \|\mathbf{v} - (\mathbf{v}^n + \mathbf{g}\Delta t)\|_\mathbf{M}^2 + \sum_q V_q^0 \Psi(\mathbf{F}_q). \tag{3}$$

This formulation is more favored because with line search methods, convergence to a local minimum of Equation 3 can be guaranteed even when simulating challenging cases with stiff materials or large time step sizes. After solving for the velocity $\mathbf{v}^{n+1}$, FEM directly updates mesh nodal positions as $\mathbf{x}^{n+1} = \mathbf{x}^n + \Delta t \mathbf{v}^{n+1}$, while for MPM, the velocity on the grid node is interpolated to particle locations for particle advection. The background grid is reset at the beginning of each time step, which allows MPM to benefit from the conveniences of a regular grid and a mesh-free formulation at the cost of some accuracy loss due to the transfers between the grid and particles.

## 3.2 Return Mapping for Plasticity

With plasticity, objects can undergo both plastic and elastic deformations, and the deformation gradient at the current time step can be decomposed as

$$\mathbf{F}^{n+1} = \mathbf{F}^{E,n+1}\mathbf{F}^{P,n+1} \tag{4}$$

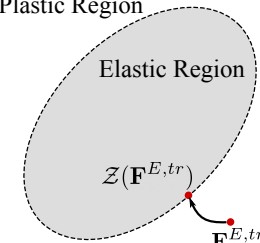

Plastic Region

Elastic Region

$\mathcal{Z}(\mathbf{F}^{E,tr})$

$\mathbf{F}^{E,tr}$

Figure 1: An illustration of a return mapping.

based on the finite strain theory. Here $\mathbf{F}^{P,n+1}$ encodes the permanent plastic deformation of the rest shape, and $\mathbf{F}^{E,n+1}$ is the elastic deformation that results in effective elastic forces. In theory, $\mathbf{F}^{E,n+1}$ is constrained within certain elastic regions. Computation-wise, an elastic predictor $\mathbf{F}^{E,tr} = \mathbf{F}^{n+1}(\mathbf{F}^{P,n})^{-1}$ can be computed first by assuming $\mathbf{F}^{P,n+1} = \mathbf{F}^{P,n}$. If $\mathbf{F}^{E,tr}$ is outside the elastic region, it will be projected back onto the boundary of the region to obtain $\mathbf{F}^{E,n+1} = \mathcal{Z}(\mathbf{F}^{E,tr})$ (Figure 1). This projection $\mathcal{Z}$ is called a *return mapping*. Within this framework, the implicit elastoplastic nodal force can be computed as [34]

$$\mathbf{f}_i^{n+1} = -\sum_q V_q^0 \boldsymbol{\tau}(\mathcal{Z}(\mathbf{F}_q^{E,tr}))\mathbf{F}_q^{E,tr^{-\top}}\mathbf{F}^{P,n^{-\top}}\nabla w_{iq} \tag{5}$$

where $\boldsymbol{\tau}(\mathbf{F}) = \mathbf{P}(\mathbf{F})\mathbf{F}^{\top}$ is the Kirchoff stress. The above forces are integrable only if $\boldsymbol{\tau}(\mathcal{Z}(\mathbf{F}))\mathbf{F}^{-\top}$ can be represented as the gradient of some energy function:

$$\frac{\partial \Psi}{\partial \mathbf{F}} = \boldsymbol{\tau}(\mathcal{Z}(\mathbf{F}))\mathbf{F}^{-\top}. \tag{6}$$

Most combinations of elastic constitutive models and plastic return mappings do not satisfy this integrability condition because the Jacobian field of the right-hand side is asymmetrical. Note that directly feeding $\mathcal{Z}(\mathbf{F})$ into an elastic potential does not form a potential energy for the elastoplastic forces defined in Equation 5. [34] only found one specific combination such that an elastoplastic potential energy exists. Thus, it remains challenging to simulate versatile plastic behaviors with optimization time integrators and achieve robust performance.

## 4 PlasticityNet

We propose PlasticityNet, a neural network-based elastoplastic model that finds a family of local potential energies whose negative gradients can approximate the elastoplastic forces within a small neighborhood so that plasticity can be conveniently simulated using optimization time integrators. The model architecture is illustrate in Figure 2. Specifically, instead of finding a global energy function $\Psi(\mathbf{F})$, we search for an energy $\Psi(\mathbf{F}, \mathbf{F}_0)$, parameterized by $\mathbf{F}_0$, such that

$$\frac{\partial \Psi}{\partial \mathbf{F}}(\mathbf{F}, \mathbf{F}_0)|_{\mathbf{F}=\mathbf{F}_0} = \boldsymbol{\tau}(\mathcal{Z}(\mathbf{F}_0))\mathbf{F}_0^{-\top}, \quad \text{and} \quad \frac{\partial \Psi}{\partial \mathbf{F}}(\mathbf{F}, \mathbf{F}_0) \approx \boldsymbol{\tau}(\mathcal{Z}(\mathbf{F}))\mathbf{F}^{-\top}. \tag{7}$$

To exactly enforce the first equality, we propose the following linear correction:

$$\Psi_\theta(\mathbf{F}, \mathbf{F}_0) = \mathcal{NN}_\theta(\mathbf{F}, \mathbf{F}_0) - (\nabla_\mathbf{F}\mathcal{NN}_\theta(\mathbf{F}_0, \mathbf{F}_0) - \boldsymbol{\tau}(\mathcal{Z}(\mathbf{F}_0))\mathbf{F}_0^{-\top}) \odot \mathbf{F}. \tag{8}$$

Here $\mathbf{A} \odot \mathbf{B} = A_{ij}B_{ij} = \text{tr}(\mathbf{A}^{\top}\mathbf{B})$ is the matrix inner product. It can be verified that $\frac{\partial \Psi_\theta}{\partial \mathbf{F}}(\mathbf{F}, \mathbf{F}_0)|_{\mathbf{F}=\mathbf{F}_0} = \boldsymbol{\tau}(\mathcal{Z}(\mathbf{F}_0))\mathbf{F}_0^{-\top}$.

Then we only need to focus on the approximation part in Equation 7. We design the training loss function for our neural network as

$$\mathcal{L}(\theta) = \mathbb{E}_{\mathbf{F}_0}\mathbb{E}_\mathbf{F}\left\|\frac{\partial \Psi_\theta}{\partial \mathbf{F}}(\mathbf{F}, \mathbf{F}_0) - \boldsymbol{\tau}(\mathcal{Z}(\mathbf{F}))\mathbf{F}^{-\top}\right\|_F^2. \tag{9}$$

During training, $\mathbf{F}$ is only sampled near $\mathbf{F}_0$. Please refer to Section 5.1 for details.

### 4.1 Hardening of Plasticity

Hardening effects are widely observed in metals and snow. With hardening, the elastic region will expand by a certain amount whenever $\mathbf{F}^{E,tr}$ falls in the plastic region. To account for hardening, the return mapping $\mathcal{Z}(\mathbf{F}, h)$ and the energy $\Psi_\theta(\mathbf{F}, \mathbf{F}_0, h)$ will depend on an extra hardening state $h$, which controls the shape of the elastic region. This hardening state is a function of $\mathbf{F}$. However, to maintain integrability with respect to $\mathbf{F}$, we approximately update $h$ based on $\mathbf{F}_0$, which is assumed to be close to $\mathbf{F}$.

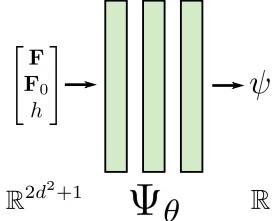

Figure 2: An overview of PlasticityNet. It is a map from $\mathbb{R}^{2d^2+1}$ to $\mathbb{R}$.

### 4.2 Optimization Time Integration with PlasticityNet

**Fixed-Point Iteration** The gradient of our learned elastoplastic potential energy $\Psi_\theta(\mathbf{F}, \mathbf{F}_0)$ only approximates the effective stresses locally near $\mathbf{F}_0$. To approach the accurate solution of Equation 1 with elastoplastic forces, we apply a fixed-point iteration on $\mathbf{F}_0$ to let it converge to $\mathbf{F}^{n+1}$. Specifically, we solve a sequence of optimization problems

$$\mathbf{v}^{n+1,j+1} = \operatorname{argmin}_{\mathbf{v}} \frac{1}{2}\|\mathbf{v} - (\mathbf{v}^n + \mathbf{g}\Delta t)\|_{\mathbf{M}} + \sum_q V_q^0 \Psi_\theta(\mathbf{F}_q, \mathbf{F}_{0,q}^j, h_q^j), \quad \text{for } j = 0, 1, 2, ..., \quad (10)$$

treating the concatenated deformation gradients $\mathbf{F}_0^j$ and hardening states $\mathbf{h}$ as constants, which are only updated before each optimization as $\mathbf{F}_0^j = \mathbf{F}(\mathbf{v}^{n+1,j})$ and $\mathbf{h} = \mathbf{h}(\mathbf{F}_0)$. At convergence, we will obtain the true solution of Equation 1. In practice, a few number of fixed-point iterations can already generate high-quality results.

**Stability Regularizer** We augment our learned potential with an extra quadratic regularizer to stabilize the simulation especially when the material is stiff or the time step size is large:

$$\Psi_\theta(\mathbf{F}, \mathbf{F}_0) = \mathcal{NN}_\theta(\mathbf{F}, \mathbf{F}_0) - (\nabla_{\mathbf{F}}\mathcal{NN}_\theta(\mathbf{F}_0, \mathbf{F}_0) - \boldsymbol{\tau}(\mathcal{Z}(\mathbf{F}_0))\mathbf{F}_0^{-\top}) \odot \mathbf{F} + \frac{1}{2}\mu\|\mathbf{F} - \mathbf{F}_0\|_F^2. \quad (11)$$

Here $\mu$ is the shear modulus of the material that $\Psi_\theta$ is learning. Note that this extra term is added after the model is trained instead of during the training. This extra term does not change the gradient at $\mathbf{F}_0$, so it will not change the fixed point of Procedure 10. Please see Section 5.3 for a comparison between simulations with and without this regularizer.

### 4.3 Learning Volume-Preserving Return Mapping

The return mapping $\mathcal{Z}$ required by PlasticityNet can be either given analytically or learned. Note that with different combinations of many practical elasticity and plasticity models, the return mapping may not have a closed-form solution, and the projection can only be performed by solving a nonlinear system of equations.

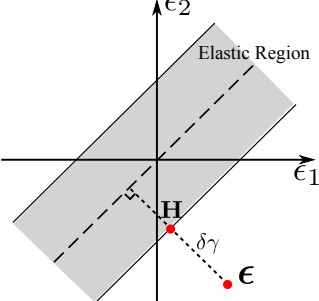

Figure 3: Volume-preserving projection.

Here we provide a simple approach to learn a volume-preserving return mapping, which ensures that $\det(\mathcal{Z}(\mathbf{F})) = \det(\mathbf{F})$. For isotropic materials, the projection can be performed in the diagonal space, i.e., with $\mathbf{F} = \mathbf{U}\operatorname{Diag}(\boldsymbol{\Sigma})\mathbf{V}^\top$ being the singular value decomposition of $\mathbf{F}$; the projection is only needed for $\boldsymbol{\Sigma}$. In the diagonal space, a volume-preserving path is a straight line in the Hencky strain (defined as $\boldsymbol{\epsilon} = \log(\boldsymbol{\Sigma})$) space, which is perpendicular to the diagonal line. The direction of the projection path is $\hat{\boldsymbol{\epsilon}} = \boldsymbol{\epsilon} - \operatorname{sum}(\boldsymbol{\epsilon})\mathbf{1}$. The volume-preserving projection in the Hencky strain can be unified by $\mathbf{H} = \boldsymbol{\epsilon} - \delta\gamma\frac{\hat{\boldsymbol{\epsilon}}}{\|\hat{\boldsymbol{\epsilon}}\|}$ for some $\delta\gamma$, with $\mathcal{Z}^\Sigma(\boldsymbol{\Sigma}) = \exp(\mathbf{H})$ and $\mathcal{Z}(\mathbf{F}) = \mathbf{U}\operatorname{Diag}(\mathcal{Z}^\Sigma)\mathbf{V}^\top$. An illustration is shown in Figure 3.

The elastic region is usually represented by an implicit function $y(\boldsymbol{\Sigma}) \leq 0$. We can use a neural network to predict $\delta\gamma$, where the training leverages the differentiability of the implicit representation for the elastic region boundary. The volume-preserving path usually has two intersections with the

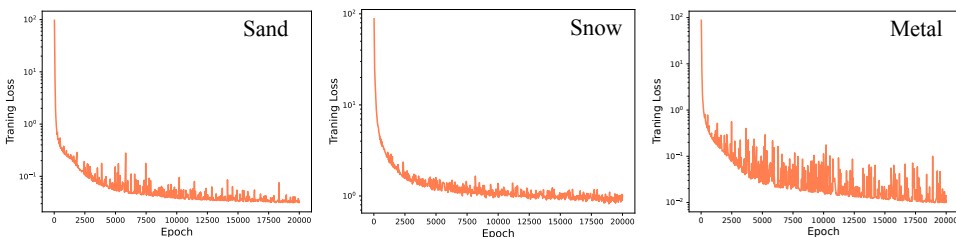

Figure 4: Training losses of our 2D models.

elastic region boundary. To eliminate this ambiguity, we clamp the output of the neural network with a maximum $\|\hat{\boldsymbol{\epsilon}}\|$. We define our neural-network-based return mapping on the diagonal space as:

$$\delta\gamma_\theta(\boldsymbol{\Sigma}) = \min\{\mathcal{NN}_\theta(\boldsymbol{\Sigma}), \|\hat{\boldsymbol{\epsilon}}\|\}, \quad \mathcal{Z}_\theta^\Sigma(\boldsymbol{\Sigma}) = \begin{cases} \exp(\boldsymbol{\epsilon} - \delta\gamma_\theta \frac{\hat{\boldsymbol{\epsilon}}}{\|\hat{\boldsymbol{\epsilon}}\|}), & y(\boldsymbol{\Sigma}) > 0, \\ \boldsymbol{\Sigma}, & y(\boldsymbol{\Sigma}) \leq 0. \end{cases} \tag{12}$$

The training loss function for a single $\boldsymbol{\Sigma}$ is defined as

$$\mathcal{L}(\boldsymbol{\Sigma}; \theta) = \begin{cases} y(\mathcal{Z}_\theta^\Sigma(\boldsymbol{\Sigma}))^2 + \max\{\delta\gamma_\theta(\boldsymbol{\Sigma}) - \|\hat{\boldsymbol{\epsilon}}\|, 0\}, & y(\boldsymbol{\Sigma}) > 0 \\ 0, & y(\boldsymbol{\Sigma}) \leq 0 \end{cases} \tag{13}$$

Here, the first term is to pull the points outside the elastic region back onto the boundary. The second term is to avoid these points to be always projected onto the diagonal due to the clamping in $\delta\gamma_\theta$. To account for hardening, we only need to let the $\delta\gamma$ network accept an extra hardening state variable $h$: $\delta\gamma_\theta(\boldsymbol{\Sigma}, h) = \min\{\mathcal{NN}_\theta(\Sigma, h), \|\hat{\boldsymbol{\epsilon}}\|\}$. The learned return mapping is then ready to be used by our PlasticityNet.

## 5 Experiments

We show examples to demonstrate the capability of our PlasticityNet in learning versatile plasticity models. Our physical simulators are implemented using C++, and we applied PyTorch to learn the potential energies, which are then loaded into our simulators with TorchScript. All our potential energies are trained as multilayer perceptrons using the Adam optimizer [22] on a single Nvidia RTX 3090 GPU. Please see Appendix A.1 for more training details. All ground-truth data are generated using standard explicit time integration with analytical plasticity returning mapping under small time step sizes for stability. With our PlasticityNet, we can robustly simulate elastoplastic behaviors with much larger time step sizes using optimization time integrators.

### 5.1 Training

The training of PlasticityNet only requires the return mapping (either given analytically or pre-trained) for the plasticity model and the Kirchhoff stress for the underlying elasticity model. There is no need for extra labeled data. At each epoch, we will sample a new batch of $(\mathbf{F}, \mathbf{F}_0, \mathbf{h})$. The sampling of deformation gradients is based on its singular value decomposition $\mathbf{F} = \mathbf{U}\,\mathrm{Diag}(\boldsymbol{\Sigma})\mathbf{V}^\top$, with $\mathbf{U}, \mathbf{V}$ being two rotation matrices. To sample $\mathbf{F}$ and $\mathbf{F}_0$ so that their singular values are close to each other, we set $\mathbf{F}_0 = \mathbf{R}_1\,\mathrm{Diag}(e^{\boldsymbol{\epsilon}})\mathbf{R}_2$ and $\mathbf{F} = \mathbf{R}_3\,\mathrm{Diag}(e^{\boldsymbol{\epsilon}+\delta\boldsymbol{\epsilon}})\mathbf{R}_4$, where $\boldsymbol{\epsilon}$ is a randomly sampled vector, $\delta\boldsymbol{\epsilon}$ is a random perturbation, and $\mathbf{R}_i$'s are randomly sampled rotation matrices. The hardening state is sampled uniformly from an appropriate range depending on the plasticity model. Please see Appendix A.2 for definitions of hardening states and their range selections. In this work, we uniformly sample $\boldsymbol{\epsilon}$ from $[-1, 1]^d$, $\delta\boldsymbol{\epsilon}$ from $[-0.1, 0.1]^d$ for sand plasticity and metal plasticity, and $[-0.2, 0.2]^d$ for the snow plasticity. The training loss curves of our 2D models are shown in Figure 4.

### 5.2 Testing on 2D Simulations

In this section, explicit time integrators are used to generate the ground-truth data for the validation of the optimization time integrators with PlasticityNet on multiple 2D experiments. The quantitative comparisons are plotted in Figure 5. We additionally include the computational costs in Table 1. We

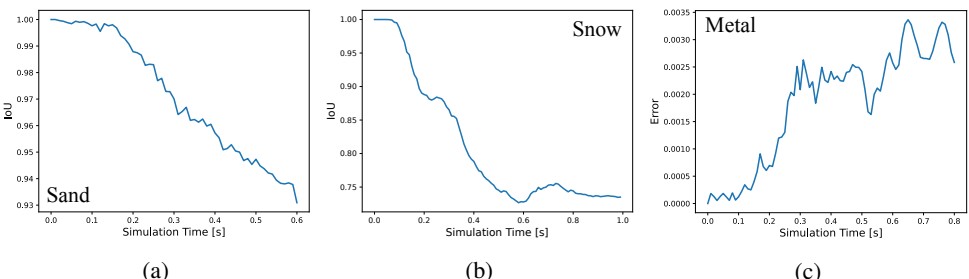

|  | (a) | (b) | (c) |

Figure 5: **(a) (b)** The intersection-over-union (IOU) [17] measure between the ground truth and our results. The IOUs are computed using the mass distributions on the MPM grid. **(c)** The average FEM nodal position difference. Note that the bounding box of the 2D metal frame is $0.1m \times 0.18m$.

Table 1: Computational costs of 2D experiments.

| Experiment | Ours | | Explicit | |
| | Time step (s) | s/frame | Time step (s) | s/frame |
| --- | --- | --- | --- | --- |
| Sand | 1e-3 | 12.58 | 1e-5 | **6.20** |
| Snow | 1e-3 | 35.56 | 1e-5 | **6.78** |
| Von-Mises Metal | 1e-2 | **1.08** | 1e-5 | 5.39 |
| Neohookean Metal | 1e-2 | **1.03** | 1e-5 | 7.88 |
| MPM-FEM Coupling | 1e-3 | **38.90** | 1e-6 | 184.58 |

remark that the main objective of our work is not to surpass the performance of the existing simulation of every constitutive model, but to provide a methodology that enables the usage of implicit plasticity in an optimization time integration framework.

**Sand Plasticity** We start by learning the elastoplastic model of dry sand (Figure 6). The model combination is St. Venant-Kirchhoff (StVK) elasticity, and the closed-form Drucker Prager plasticity return mapping [23] (See Appendix A.2.1). In this example, we simulate a column of sand falling onto the ground under gravity with MPM. Our method generates visually identical results compared to the ground truth, both with the same time step size and a $100\times$ larger time step size. The quantitative comparison between our results and the ground truth is shown in Figure 5a. Note that there is no hardening mechanism in this plasticity model, so our PlasticityNet does not need the hardening state in its input.

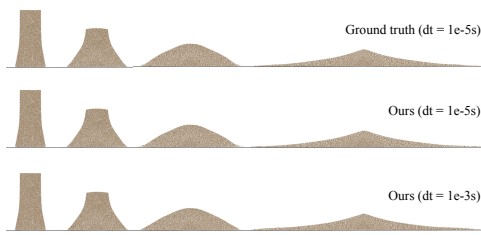

Figure 6: Sand plasticity.

**Snow Plasticity with Hardening** Snow is an elastoplastic material that can become stiffer under compression. Essentially, this is the effect of hardening where its elastic region get expanded. The variation in the stiffness across the snow body makes it easily fracture. Here we simulate a snowball hitting the ground in the MPM simulator (Figure 7). We use Neo-Hookean elasticity with the closed-form non-associative Cam-Clay plasticity return mapping [15] (See Appendix A.2.2). Our method generates similar results compared to the ground truth when using the same time step size. The quantitative comparison of our results and the ground truth is shown in Figure 5b. Our framework remains stable even under

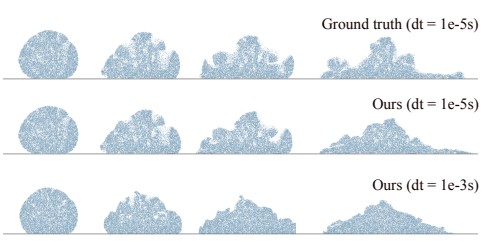

Figure 7: Snow plasticity with hardening.

much larger time step sizes. However, more numerical damping artifacts are introduced as the time step size increases, which results in slightly different behaviors compared to the ground truth.

**Metal Plasticity with Hardening** Metal is another common plastic material with hardening. In this example, we train PlasticityNet to learn metal plasticity with the StVK elasticity and the closed-form von-Mises plasticity return mapping [39]. (See Appendix A.2.3) We simulate a metal frame compressed by a rigid plate in the FEM simulator (Figure 8), where the Incremental Potential Contact (IPC) [29] is used to handle the frictional contact between the solids. When we run the explicit time integration to generate ground truth, we have to decrease Young's modulus to enable using large enough time step sizes so that the simulation can be finished in practical time. Our method with the original setting generates visually identical results using a much large time step size. The quantitative comparison of our result and the ground truth is shown in Figure 5c.

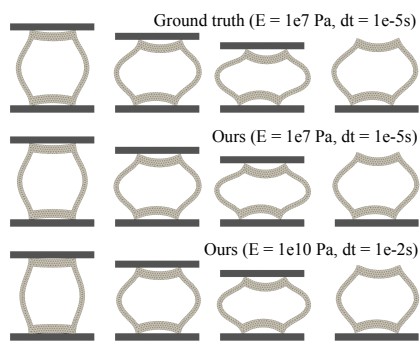

Figure 8: Metal plasticity with hardening.

**Metal Plasticity Return Mapping** Here we show an example simulated using PlasticityNet with a learned von-Mises plasticity return mapping. The underlying elasticity is neo-Hookean, instead of the StVK model in the last example (See Appendix A.2.4). Note that for Neo-Hookean material, there is no closed-form solution available for the von-Mises return mapping. In this case, a nonlinear

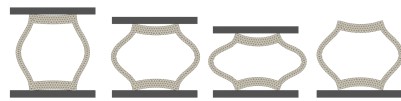

Figure 9: Learned metal plasticity return mapping with neo-Hookean elasticity.

optimization problem will need to be solved to perform the return mapping for every element/particle in every time step, which could severely slow down the standard explicit time integration. Using the same parameters as the metal compression experiments above, we show that PlasticityNet with learned plasticity return mapping under neo-Hookean elasticity can generate qualitatively similar results (Figure 9) to those from PlasticityNet with closed-form return mapping under the StVK elasticity.

**MPM-FEM Coupling** PlasticityNet enables the simulation of plastic materials in the MPM-FEM coupling framework BFEMP [33], where only pure elasticity was supported. When simulating with explicit BFEMP, the time step size required by stability is the minimum between MPM step size upperbound and FEM step size upperbound. Here we show an example where a stiff FEM elastic body falls onto MPM sand (Figure 10), where the implicit BFEMP can use a time step size 1000x larger than the explicit BFEMP and achieves an approximately 5x

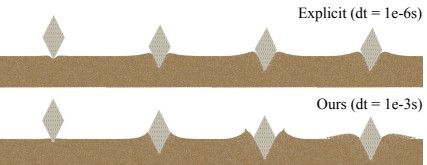

Figure 10: Two-way coupling between FEM elasticity and MPM sand plasticity.

speedup in wall-clock time. We also remark that when the time step size is small (as is required to keep the explicit time integration stable in this case), MPM suffers from excessive numerical damping due to the significant amount of particle-grid transfers. This is a known issue of explicit MPM simulations.

**Different Energy Representations** Here we include some different energy representations we investigated (Figure 11), whose inaccurate results motivated us to develop our final representation Equation 8. These experiments are all conducted on the 2D sand column collapse example. The first straightforward idea is to find a globally defined neural energy function $\Psi(\mathbf{F}) = \Psi_\theta(\mathbf{F})$ that solves Equation 6, where $\theta$ is the parameter of the neural network. Note that it is theoretically unachievable to train a global potential energy function because the right hand side of Equation 6 is not integrable in the plastic region. But it is still worth trying to explore an approximation by minimizing $\mathcal{L}(\theta) = \mathbb{E}_{\mathbf{F}} \left\| \frac{\partial \Psi_\theta}{\partial \mathbf{F}}(\mathbf{F}) - \boldsymbol{\tau}(\mathcal{Z}(\mathbf{F}))\mathbf{F}^{-\top} \right\|_F^2$. However, the experiment shows that this formulation makes the sand column behave like an elastic body. It is also noteworthy that the sand column cannot even maintain the rest shape at the first frame: it erroneously shrinks suddenly and jumps off the ground. Additional insight is provided by realizing that a linear correction is necessary to exactly vanish stress when the deformation gradient is the identity; so we experiment with $\Psi(\mathbf{F}) = \Psi_\theta(\mathbf{F}) - \nabla_{\mathbf{F}}\Psi_\theta(\mathbf{I})$.

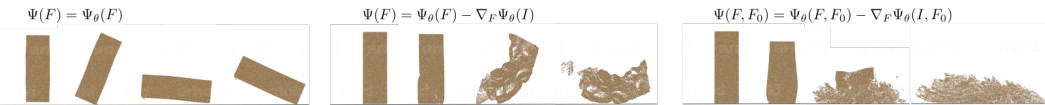

Figure 11: Ablation studies on different energy representations.

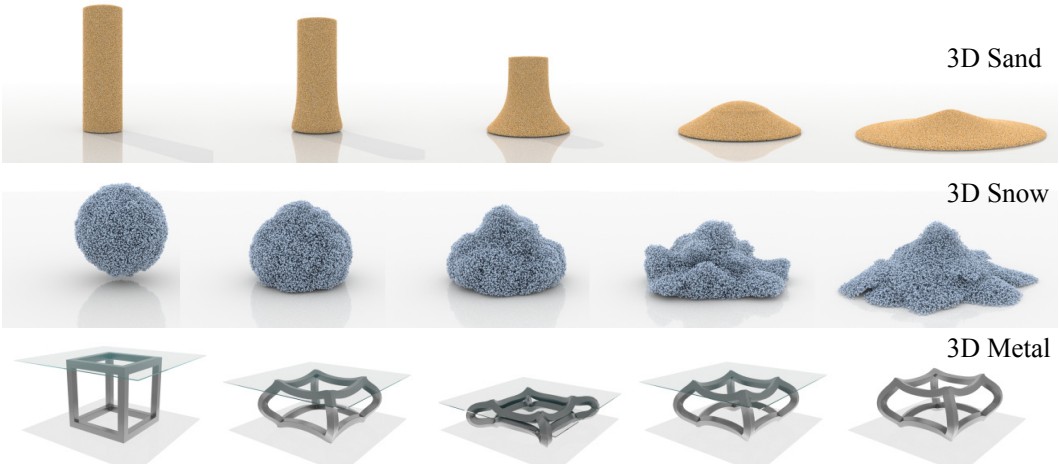

Figure 13: 3D simulations with sand plasticity, snow palsticity and metal plasticity.

This formulation unfortunately also leads to an insufficient capture of plasticity, giving an elastic and visually distinct incorrect result. These observations motivate us to investigate a family of potential energies to solve Equation 6 locally. We first use $\Psi(\mathbf{F}, \mathbf{F}_0) = \Psi_\theta(\mathbf{F}, \mathbf{F}_0) - \nabla_\mathbf{F}\Psi_\theta(\mathbf{I}, \mathbf{F}_0)$ and train with the loss function in Equation 9. The simulation captures certain plastic behaviors when the deformation is small, but the result quickly deviates from the ground truth when the deformation becomes larger. Finally, we come up with Equation 8 to achieve the nice results in Figure 6.

### 5.3 Ablation Studies

**Stability Regularizer**   As an ablation study for the stability regularizer in Equation 11, we compare the simulations with and without the regularizer on two 2D examples (Figure 12). Without the regularizer, the metal frame can not even stay in its original rest configuration after the first time step. In the sand example, particles in the highlighted regions tend to separate from the sand column in a non-physical manner. These demonstrate that our regularizer significantly improves the stability of the simulation.

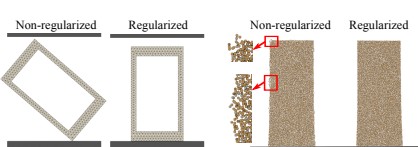

Figure 12: The regularizer significantly improves the stability of the simulation.

### 5.4 Testing on 3D Simulations

Extending PlasticityNet to support 3D simulation is straightforward. We only need to increase the dimension of the inputs to the PlasticityNet. To improve the expressiveness of the network, we also increase the dimension of hidden variables. Here we demonstrate the 3D versions of our 2D examples with similar physical parameters in Figure 13: 3D sand plasticity, 3D snow plasticity, and 3D metal plasticity. The 3D metal is simulated with $\Delta t = 10^{-2}$, and for sand and snow, we use $\Delta t = 10^{-3}s$ to satisfy the CFL condition [8] in MPM, preventing the particles from traveling farther than the grid cell spacing in a single time step.

## 6   Conclusion

We proposed PlasticityNet, a neural network-based elastoplastic model learning framework that is agnostic to spatial discretizations. PlasticityNet represents the elastoplastic forces as the positional

gradients of learned potential energies, so that optimization time integration could be applied to achieve robust and efficient simulation at large time step sizes. We demonstrated that low-level components in traditional physical simulation frameworks can be substituted with neural networks to obtain desired numerical properties that benefit the computation. Notably, this also avoids tedious analytical derivations or expensive nonlinear root-findings without significantly sacrificing the accuracy. We believe our work can inspire more research that applies machine learning to physical simulation in the bottom-up style, maintaining fundamental physical properties and applicability to general scenarios.

**Limitations and Future Work** There are several limitations of our framework. **(1)** We cannot guarantee our fixed-point iteration will converge for arbitrary scenes. It is theoretically valuable to explore under what conditions the fixed-point can converge and what loss functions can accelerate the convergence. **(2)** Although the regularizer added during the simulation improves the stability of the simulation without changing the solution at convergence, it may introduce some artificial viscosity because the regularized energy penalizes deformations away from $\mathbf{F}_0$. Running more fixed-point iterations can alleviate this issue. It will also be interesting to explore adaptive weighting mechanisms for the regularizer, or convert this soft regularizer into a hard constraint. **(3)** We do not consider the Hessian of the learned plastic energy in our training. Since we use second-order methods to perform optimization time integration, the properties of the Hessian matrices may have an impact on the convergence of the optimization method. Although the Jacobian matrices of the target gradients are asymmetric, it may be helpful if the Hessian of our learned elastoplastic energy can approximate them so that the stiffness of the material can be more accurately resolved. **(4)** Principled physical assumptions of the learned potential energies by PlasticityNet, such as lower-boundedness and convexity, are not enforced. It is interesting to explore whether enforcing these energy properties would positively influence the convergence of the optimizations and fixed-point iterations. **(5)** A trained PlasticityNet can be directly re-scaled to accommodate a different Young's modulus, but it needs to be re-trained for materials with different Poisson's ratio or plasticity parameters. It is an important future work to let our model more easily generalize to different parameters. For example, these parameters can become extra inputs to the neural network. The generalized energy can also be integrated into differentiable simulators [16, 43] to solve many inverse problems [38, 36, 32].

## Acknowledgments and Disclosure of Funding

We would like to thank Pingying Chen for narrating the supplemental video. We would also like to thank the anonymous reviewers for their valuable comments. This work has been supported in part by NSF CAREER 2153851, CCF-2153863, ECCS-2023780, IIS-2011471, IIS-2016414, IIS-2006570, DOE ORNL contract 4000171342.

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
