# OpenReview forum: "PlasticityNet: Learning to Simulate Metal, Sand, and Snow for Optimization Time Integration"
_NeurIPS.cc/2022/Conference — NeurIPS 2022 Accept_

### Official Review · Reviewer_8rPt · 2022-07-09

**Rating:** 6
**Confidence:** 4
**Soundness:** 3 good
**Presentation:** 3 good
**Contribution:** 2 fair

**Summary:**

The authors proposed a learning-based model to simulate plasticity effects. The method is based on optimization based time integration and trains a network to predict the energy given the deformation gradients and the hardening states. Results show that the predicted energy can successfully guide the simulation to behave similarly as ground truth ones.


**Questions:**

See above.

**Limitations:**

No. See above.

**Strengths And Weaknesses:**

Strengths:
 - The network only tries to learn the energy mapping for a certain point, which provides the ability to scale the method to arbitrary scenes.
 - Using neural network to predict the energy can enable larger step time, which in turn speed up the simulation.

Weaknesses:
 - Missing references:
   - Hu, Y., Anderson, L., Li, T. M., Sun, Q., Carr, N., Ragan-Kelley, J., & Durand, F. (2019, September). DiffTaichi: Differentiable Programming for Physical Simulation. In International Conference on Learning Representations.
   - Qiao, Y., Liang, J., Koltun, V., & Lin, M. (2021). Differentiable simulation of soft multi-body systems. Advances in Neural Information Processing Systems, 34, 17123-17135.
   These two papers combines machine learning with simulation via differentiating the simulation as a whole which can enable efficient inverse problem solving as well. I suggest the authors discuss this direction and if possible, compare with DiffTaichi since it also uses MPM.
 - It is not clear how much performance gain there is after using the learned model. The step time is indeed 100x larger (from 1e-5 to 1e-3, as mentioned in the text), but to achieve convergence, the authors adopted a fixed point iteration which is not required in traditional simulations. *A few number* of iterations as mentioned by the authors are not accurate enough to evaluate the strengths of this approach. I would like to see performance tables or figures that shows significant improvement with the proposed approach so that it can be strongly supported.
 - It seems the network has to be trained all over again if the material parameters are different, which limits the generalization ability of this approach. While it is a common open problem for all learning-based simulations, I suggest the authors discuss and provide insights towards solving it in the limitation and future work section.
 - It is not clear how the 100x step time is achieved. Why 100x but not 1000x? Is there any theorectical hypothesis on how much at most a learning based model can improve? Does the error goes up with the step time, and how is the trend like?

---

> ### Author Response · Authors · 2022-08-01
> **Reply to Reviewer 8rPt**
>
> Thanks for your detailed and informative reviews.
>
> **Here are our answers to your comments on our weakness:**
>
> 1. The two references focus on using differentiable simulators to solve inverse problems, which is out of the scope of our paper. But solving inverse problems is an important future work of our paper, and we cite them in future work section.
> 2. The computational time statistics are included in the revised paper (Table 1). Please also refer to our reply to all reviewers for performance discussions.
> 3. Our PlasticityNet is invariant under different Young’s modulus. But we need to train a new network if Poisson’s ratio or plasticity parameters change. It is indeed an interesting future work to include different plasticity parameters in one network. For example, let plasticity parameters be the inputs to the neural network as well. Thanks for pointing it out.
> 4. The time step size of MPM is restricted by the [CFL condition](https://en.wikipedia.org/wiki/Courant-Friedrichs-Lewy_condition#The_one-dimensional_case), i.e., a particle cannot move more than dx (the grid spacing) in a single time step. In our 2D sand column experiments, dx = 2e-3 m, the maximal velocity is close to 2 m/s. So the maximal permitted time step size is around dt = 1e-3 s. In addition, the error of numerical solutions to PDE will increase as dt becomes larger. The global error for backward Euler is C(T)dt, and the constant C(T) will increase as the total simulation time T increases.

---

> > ### Comment · Reviewer_8rPt · 2022-08-08
> > **Updates**
> >
> > The authors have addressed my concerns thoroughly. After reading all comments from other reviewers, I decided to change my score to 6.

---

### Official Review · Reviewer_LXcT · 2022-07-10

**Rating:** 6
**Confidence:** 3
**Soundness:** 2 fair
**Presentation:** 3 good
**Contribution:** 2 fair

**Summary:**

This paper presents a neural network model to learn a plastic material’s energy function in elastoplastic simulation. The paper argues that constructing such an energy model enables optimization-based solvers for the time integration problem, which is more advantageous than solving the force equilibrium directly. Additionally, the paper introduces a few useful extensions on top of the network model: hardening effects, stability regularizer, and volume-preserving return mapping. The paper presents various 2D and 3D examples (sand, snow, and metal) to demonstrate the value of the proposed network.

**Questions:**

1. For solving the time integration, this paper seems to favor energy minimization (Eqn. 3) instead of solving the governing equation (Eqn. 1). However, I am under the impression that both approaches have been equally popular in deformable-body simulation, and the argument in lines 116-118 doesn’t fully convince me because we could also solve Eqn. 1 using Newton’s method plus line search/other safeguards. Maybe I am unaware of some critical reference papers here. Could you please comment more on this?

2. On a related note, I am not sure I get what lines 138-140 intend to say. To me, directly feeding Z(F) to the elastic potential (followed by solving the nonlinear root-finding or energy minimization problem) seems to be a legit solution. Could you clarify this?

3. I really appreciate the magic equation 8. Could you tell me more about how you derived it?

4. Eqn 11: why is shear modulus used as the weight of the regularizer? Is there physical intuition behind this choice?

5. One general concern that I always have with such network methods is that they don’t seem to guarantee physical plausibility. For example, a reasonable energy potential is expected to be bounded below, which is easy to guarantee for an analytical energy potential function but could be difficult to enforce for a network model (The discussion on Hessians in the limitation section might be somewhat relevant to this point). Could you give me a strong motivation for why we should appreciate these network models?

6. I am confused by the experiments that compare ground-truth simulation generated using explicit time integration with your approach that solves implicit time integration. Why not use implicit time integration in both settings?

7. I also didn’t find statistics about the wall clock time cost. I get that having the network to model the energy potential unlocks a larger time step than explicit methods, but doesn’t each implicit time step cost more wall-clock time? Therefore, what is the final time cost for generating each row in Fig. 5?

8. For Fig 7 row 2 and row 3: not sure I understand why adjusting both time step size and Young’s modulus between these two rows.

9. I also didn’t find a discussion about the generalization of the trained network model. Actually, I am not sure how generalization/training data/test data should be defined for this model. I am raising this concern because network models typically over-parametrize physics models (physics laws often only have a few parameters while networks can easily have thousands or more).


**Limitations:**

The limitation section looks good to me.

**Strengths And Weaknesses:**

Strength: this paper discussed a serious physics simulation problem. The mathematical and physical background is fairly solid. Implementing MPM, FEM, and IPC are nontrivial, for which the authors deserve some credits if they coded everything from scratch. The 2D and 3D demos are also visually pleasing.

Weaknesses: I feel the paper still lacks a deeper insight behind the introduction of the network model. I also think the evaluation section misses some critical points.

Overall, I think this paper is very interesting and promising, but I am afraid it is not there yet. My initial score is rejection, but I am happy to re-evaluate the paper if the authors can address my questions.

---

> ### Author Response · Authors · 2022-08-01
> **Reply to Reviewer LXcT**
>
> Thanks for your detailed and informative reviews.
>
> **Here are our answers to your questions:**
> 1. Directly solving Eqn 1 requires solving a nonlinear system with root-finding Newton method, which has super-linear convergence only when the starting point is sufficiently close to the optima (when dt is small or materials are soft). In practice, root-finding Newton method may even diverge if the starting point is too far away. Instead, optimization time integration can guarantee energy decrease in every iteration, so that a local minimum is guaranteed to be found, and the solution also satisfies Eqn 1. More detailed discussion of the advantages of optimization time integrations over root-finding implicit integrations can be found in the paper:
>
>     *Gast, et al. 2015. Optimization integrator for large time steps. IEEE Trans Vis Comput Graph.*
>
> 2. This will be easier to clarify in a 1D case. In 1D, the deformation gradient F is a scalar. Assuming perfect plasticity, Z(F) = F_Y if F > F_Y, which means the energy outside the elastic region is constant, providing zero force. However, the force should not be zero because of elasticity.
>
> 3. As revealed by Eqn 5 and 6, the internal force is determined by the derivative of the strain energy. The logic behind Eqn 8 is that we would like to correct the derivative of the neural network at F_0. The easiest way to do this correction is to add linear functions, since their gradients are constant. More specifically, assume f(x; s) is given, where s is a parameter. We would like to find g(x; s) such that g’(x_0; s) = a. First we can eliminate the gradient of f(x; s) at x_0 by substituting its first-order Taylor expansion around x_0 from f: h(x; s) = f(x; s) - f’(x_0; s)(x - x_0). Then we have h’(x_0; s) = 0. We can add the required gradient back by adding a(x-x_0) to h. Then we have g(x; s) = f(x; s) - f’(x_0; s)(x - x_0) + a(x-x_0). Since we don’t have constraints on function values, we can drop constant terms to get g(x; s) = f(x; s) - [f’(x_0; s) - a]x, and g’(x; s) = f’(x; s) - f’(x_0; s) + a. Now we still have s as the degree of freedom to control g’(x; s) without changing g’(x_0; s) = a.
>
> 4. The choice of shear modulus is based on two considerations: 1) since shear modulus depends on Young’s modulus, using shear modulus can avoid the needs to adjust the weight for different Young’s modulus; 2) using shear modulus makes the energy correction in unit J/m^d, which is consistent with the volume integrations used in the time integration.
>
> 5. The motivation of our potential energy is that its gradient can ‌approximate elastoplastic forces so that these forces can embrace the advantages of optimization time integration. The properties of the potential energy are relatively less important. But indeed, the properties of the energy, such as lower-boundedness and convexity, affect the convergence of the optimization. It is important to control the energy property to improve the optimization cost in the future. We will add this limitation to our paper.
>
> 6. Directly solving Eqn 1 with fully implicit plastic force (Eqn 5) itself is non-trivial and worth further research. It requires solving indefinite linear systems and carefully tuning the time step size to make sure the root-find Newton converges. On the other hand, explicit time integration with plastic return mappings has been extensively explored by previous works as the elasticity and the plasticity are disentangled. So we choose the explicit simulators as the ground truth considering their easy accessibility.
>
> 7. The clock-time statistics are included in the revised paper (Table 1). Please also refer to our reply to all reviewers for performance discussions.
>
> 8. The actual Young’s modulus of metal is 1e10 Pa. Explicit simulators with this setting require a very small dt such as 1e-9, which takes days to finish a short simulation. On the other hand, the ratio between the yield stress and Young’s modulus is kept constant for the comparison, so the yielding only depends on deformations. Therefore, the results with different Young’s modulus are comparable.
>
> 9. The overall goal of PlasticityNet is to solve the small PDE system defined by Eqn 6. Like physics-informed neural networks (PINN), over-fitting is in fact not a bad thing for the purpose of finding a solution. If PlasticityNet can learn very good local approximations by over-fitting, it can accelerate the fixed-point iteration procedure in the simulation.
>
>     On the other hand, a trained PlasticityNet can be directly re-scaled to accommodate a different Young's modulus, but needs to be re-trained for different Poisson's ratios or plasticity parameters. It is an important future work to let our model more easily generalize to different parameters. For example, these parameters can become extra inputs to the neural network. We have included this point in the future work section.

---

> > ### Comment · Reviewer_LXcT · 2022-08-08
> > **Review updates**
> >
> > Thank you for answering my questions. I don't have major concerns about this paper now and have raised my score to 6.

---

### Official Review · Reviewer_RLXg · 2022-07-11

**Rating:** 7
**Confidence:** 3
**Soundness:** 3 good
**Presentation:** 3 good
**Contribution:** 3 good

**Summary:**

The authors propose a neural network-based approach, which enables the simulation of a wide range of arbitrary elasticity-plasticity combinations using time step-independent, unconditionally stable optimization-based time integrators.


**Questions:**

1. In Fig.9, why the errors of sand and snow are reduced along with simulation, but the error of metal increases.
2. How about the generalization ability to a different initial shape with the same material and a different initial shape with different material? For example, give a model trained on the sand with the initial shape of a cube (term it as 'sand cube'),  and then test it on the 'sand sphere'. Further, test the model on the 'snow sphere'.


**Limitations:**

Yes, limitations and societal impact were appropriately addressed.



**Strengths And Weaknesses:**

This paper proposes a general framework to enable the simulation of objects with a wide range of materials. The framework is novel and reliable. The paper is well-written, and the experimental setting is clear. I think this is a good paper.

---

> ### Author Response · Authors · 2022-08-01
> **Reply to Reviewer RLXg**
>
> Thanks for your detailed and informative reviews.
>
> **Here are our answers to your questions:**
> 1. We use different error measures for our MPM and FEM experiments. The error measure used in MPM is the IoU (Intersection over Union) score between MPM grid mass distributions, where a larger score implies a smaller error (1 means nearly identical). While the error measure for FEM is the L2 error between the vertex coordinates, and so smaller is better.
> 2. Our PlasticityNet is agnostic to material initial shapes. The neural network can be viewed as a strain energy density function, it can be used as a low-level constitutive model in any simulation framework with varying geometries.

---

### Official Review · Reviewer_mwtb · 2022-07-16

**Rating:** 8
**Confidence:** 4
**Soundness:** 3 good
**Presentation:** 4 excellent
**Contribution:** 4 excellent

**Summary:**

This paper proposes to learn a PlasticityNet to facilitate implicit Euler method for simulating plastic deformation.

- Given a deformation gradient, the PlasticityNet behaves like an energy function, which gradients represents the force/stress of the plastic materials. Such formulation allows the user to write the forward step as an optimization problem and enables the implicit Euler method.
- The plasticityNet only needs to be locally accurate around input deformation gradients for plastic deformation. Based on this property, the authors propose a fix-point iteration method with a regularization term that better exploits the locality to stablize the training.
- The authors conducted experiments on a wide range of materials and achieve good performance.

**Questions:**

Regarding the stability regularization, it seems that due to the locality of the PlasticityNet, we need to solve a constrained optimization problem within the pre-defined trust regions. Isn’t it better to add the hard constrain directly?

**Limitations:**

yes

**Strengths And Weaknesses:**

Strength:

- Simulating plastic material has a wide range of applications, from animation to robotics. This paper attacks the fundamental problem of plastic material simulation. It introduces a learning-based method to enable implicit Euler, which will have an impact on the field of learning-based simulation and facilitate research on other domains.
- The proposed techniques are very novel to me. Learning a Pseudo-energy function for implicit Euler is a good idea and may have potential to generalize on other fields. Besides, the idea of parameterizing the function with F_0 for local approximation and the special form that embeds physical prior both are also very interesting.
- The paper is well-written and the presentation is very clear and easy to follow.

Weakness:

- What I find missing is the discussion about the time complexity. Will the network and the fixed-point iteration introduce an additional computation burden, canceling out the benefits of the implicit Euler?
- The paper could be strengthened for readers from the learning community by adding an ablation study or baseline comparisons. For example, what would happen if we directly learn a global function without inputting F_0, or if we do not introduce the “linear correction” but learn the potential energy directly? It would be interesting to have those experiments to help understand the philosophy behind the particular design. Such ablation studies are necessary to support the method. I would also be interested if the learned network can generalize to different scenes.

---

> ### Author Response · Authors · 2022-08-01
> **Reply to Reviewer mwtb**
>
> Thanks for your detailed and informative reviews.
>
> **Here are our answers to your comments on our weakness:**
> 1. The performance speedup depends on the stiffness of the problem. Please refer to our reply to all reviewers for performance discussions.
> 2. - We have included ablation studies on different energy representations. Please refer to Section 5.3 in the revised paper for simulation results. In summary, we include experiments on three energy representations, which motivates our final representation Eqn. 8. (1) We tried to solve Eqn 6 with a global defined potential energy. The experiment shows that the sand behaves like elastic body. (2) One observation is that at least a linear correction is needed at the identity deformation gradient to exactly vanish stress when there is no deformation, otherwise, the rest shape cannot even be maintained. (3) Another baseline is to use a loss term to enforce the first part of Eqn 7 (the gradient of the local energy should equal the target). The experiment shows that the constraint is hard to be enforced with high accuracy via training. This is why we proposed to use a corrector to exactly enforce the constraint regardless of training error.
>     - For generalization, a trained PlasticityNet can be directly re-scaled to accommodate a different Young's modulus, but it needs to be re-trained for materials with different Poisson's ratio or plasticity parameters. It is an important future work to let our model more easily generalize to different parameters. For example, these parameters can become extra inputs to the neural network. We have included these in the future work section.
>
> **Here is our answer to your question:**
>
> Using hard inequality constraints for the stability regularization is an interesting direction to explore. In this way, simulation is only affected when these constraints are activated. We have included it in the future work section.
>
> We applied the soft penalty for stability regularization because 1) it is trivial to be integrated into our optimization time integration framework and it works well in all our examples; 2) if a hard inequality constraint is used, it is tricky to tune the bounds.

---

### Author Response · Authors · 2022-08-01
**Performance Discussions and the Importance of Implicit Integrators**

# PERFORMANCE RESULTS
Multiple reviewers asked about the performance. We add the timing table in the revision (Table1). Compared to explicit solvers, our method is slower on the presented MPM sand/snow examples, but much faster on FEM metal and MPM-FEM coupling[1] examples. Remarks here are:
- The speedup grows with the material stiffness. This is a common rule of thumb for implicit solvers v.s. explicit solvers.
- Implicit solvers are very advantageous for challenging cases (such as the MPM-FEM coupling), where explicit ones require non-practically small time steps.
- As shown in Fig 9 in the revision, a small time step in explicit MPM leads to excessive numerical damping and thus artificially viscous results.
# OBJECTIVE
On a high level, the objective of our work is not to use neural nets to make every example faster, particularly not for cases where explicit solvers work just fine. Our core contribution is the novel method that enables plasticity in an optimization time integrator, where traditional discoveries of analytical energies could be mathematically tedious and even impossible.
# WHY IMPLICIT
Here, we highlight a key concept in physics-based animation (PBA) and numerical methods for PDEs: why would people favor an implicit time solver if it is not always faster than an easy-to-implement explicit one? Also, why are optimization-based reformulations getting popular? These questions were rarely asked before MPM got popular since explicit solvers in FEM often require ridiculously small time steps. MPM explicit solvers permit larger time step sizes, so a good portion of MPM work is explicit. However, there is an equally large body of implicit MPM works in both graphics and computational physics.

Back to the questions, we point out that optimization time integrators have been arguably the most popular implicit solver in PBA over recent years. The benefit is way beyond the potential cost it saves with larger time steps. The important features include:
- the guarantee of stability regardless of time steps and material parameters,
- the ability to handle constraints for things like strain limiting and frictional contact,
- the support of fast optimization for fast, and even real-time simulations, and
- the support of derivative computations through the adjoint methods for inverse problems.

A large body of PBA research is dedicated to these features in both FEM and MPM communities, e.g., frictional contact FEM solids [2], real-time FEM solids [3,4,5], multigrid MPM solvers [6], MPM topology optimization [7], MPM solid-fluid coupling [8], etc. (We could go on and on.) However, an explicit solver can achieve none of these features. With an explicit solver, the time step size needs careful tuning to avoid simulation “explosion”, there is no way to guarantee the satisfaction of the constraints above, and there is no way to use fast optimizations to develop real-time simulations.
# SUMMARY
The most significant value of our method is that all the optimization integrators that have the features mentioned above can now include implicit plasticity, and what’s even better, they can use plasticity models that one cannot derive analytically (like our neo-Hookean metal example), thanks to neural nets.

We would also like to use the PINN as an analogy. PINN’s early variations are slower than almost all traditional methods for solving PDEs. NTopo [8] shows that their PINN method is much slower than the FEM method in topology optimizations. One of the most valuable contributions of PINN is that it can find differentiable solutions to PDEs using neural nets and avoid tedious mathematical derivations, which are the goals of our work as well.

In summary, we sincerely hope the reviewers could evaluate our method based on its value in enabling implicit plasticity on versatile optimization time integrators, rather than on its not necessarily better performance in some cases that explicit solvers happen to work well.

# REFERENCES
1. Li, X., et al. 2022. BFEMP: Interpenetration-free MPM–FEM coupling with barrier contact.
2. Li, M., et al. 2020. Incremental potential contact: intersection-and inversion-free, large-deformation dynamics.
3. Xian, Z., et al. 2019. A scalable Galerkin multigrid method for real-time simulation of deformable objects.
4. Bouaziz, S., et al. 2014. Projective dynamics: Fusing constraint projections for fast simulation.
5. Narain, R., et al. 2016, July. ADMM⊇ projective dynamics: fast simulation of general constitutive models.
6. Wang, X., et al. 2020. Hierarchical optimization time integration for CFL-rate MPM stepping.
7. Li, Y., et al. 2021. Lagrangian–Eulerian multidensity topology optimization with the material point method.
8. Fang, Y., et al. 2020. IQ-MPM: an interface quadrature material point method for non-sticky strongly two-way coupled nonlinear solids and fluids.
9. Zehnder, J., et al. 2021. Ntopo: Mesh-free topology optimization using implicit neural representations.

---

### Meta-Review · Area_Chair_jcWg · 2022-08-24

**Recommendation:** Accept
**Confidence:** Certain

**Metareview:**

After rebuttal, all reviewers vote to accept this submission due to its technical novelty, presentation, and wide potential applications.  The AC agrees.   Congratulations.

**Award:**

No

---

### Decision · Program_Chairs · 2022-09-14

Accept